# An association between prediction errors and risk-seeking: Theory and behavioral evidence

**Moritz Moeller** [1], **Jan Grohn** [2], **Sanjay Manohar** [1,2‡], **Rafal Bogacz** [1‡]*

**1** Nuffield Department of Clinical Neurosciences, University of Oxford, Oxford, United Kingdom,
**2** Department of Experimental Psychology, University of Oxford, Oxford, United Kingdom

☯ These authors contributed equally to this work.
‡ SM and RB also contributed equally to this work.
* rafal.bogacz@ndcn.ox.ac.uk

**Data Availability Statement:** The authors confirm that all data underlying the findings are fully available without restriction. All presented data is available at the MRC Brain Network Dynamics Data Sharing Platform https://data.mrc.ox.ac.uk/data-

## Abstract

Reward prediction errors (RPEs) and risk preferences have two things in common: both can shape decision making behavior, and both are commonly associated with dopamine. RPEs drive value learning and are thought to be represented in the phasic release of striatal dopamine. Risk preferences bias choices towards or away from uncertainty; they can be manipulated with drugs that target the dopaminergic system. Based on the common neural substrate, we hypothesize that RPEs and risk preferences are linked on the level of behavior as well. Here, we develop this hypothesis theoretically and test it empirically. First, we apply a recent theory of learning in the basal ganglia to predict how RPEs influence risk preferences. We find that positive RPEs should cause increased risk-seeking, while negative RPEs should cause risk-aversion. We then test our behavioral predictions using a novel bandit task in which value and risk vary independently across options. Critically, conditions are included where options vary in risk but are matched for value. We find that our prediction was correct: participants become more risk-seeking if choices are preceded by positive RPEs, and more risk-averse if choices are preceded by negative RPEs. These findings cannot be explained by other known effects, such as nonlinear utility curves or dynamic learning rates.

## Author summary

Many of our decisions are based on expectations. Sometimes, however, surprises happen: outcomes are not as expected. Such discrepancies between expectations and actual outcomes are called prediction errors. Our brain recognizes and uses such prediction errors to modify our expectations and make them more realistic—a process known as reinforcement learning. In particular, neurons that release the neurotransmitter dopamine show activity patterns that strongly resemble prediction errors. Interestingly, the same neurotransmitter is also known to regulate risk preferences: dopamine levels control our willingness to take risks. We theorized that, since learning signals cause dopamine release, they might change risk preferences as well. In this study, we test this hypothesis. We find that participants are more likely to make a risky choice just after they experienced an

set/behaviour-and-pupillometry-bandit-task (DOI: 10.5287/bodleian:VYZgovxVx). The code to reproduce the figures and analyses is also fully available without restrictions (https://doi.org/10. 5281/zenodo.5111690).

**Funding:** This work has been supported by Medical Research Council (MRC, mrc.ukri.org) grants MC_UU_12024/5, MC_UU_00003/1 and Biotechnology and Biological Sciences Research Council (bbsrc.ukri.org) grant B/S006338/1 held by RB, an MRC clinician scientist fellowship MR/P00878X to SGM and studentships from the MRC (MR/K501256/1 and MR/N013468/1) and St John's College to JG. The funders had no role in study design, data collection and analysis, decision to publish, or preparation of the manuscript.

**Competing interests:** The authors have declared that no competing interests exist.

outcome that was better than expected, which is precisely what our theory predicts. This suggests that dopamine signaling can be ambiguous—a learning signal can be mistaken for an impulse to take a risk.

## Introduction

Reward-guided learning in humans and animals can often be modelled simply as reducing the difference between the obtained and the expected reward—a reward prediction error. This well-established behavioral phenomenon [1] has been linked to the neurotransmitter dopamine [2]. Dopamine neurons project to brain areas relevant for reward learning, such as the striatum, the cortex and the amygdala [3,4]. Dopamine activity is known to change synaptic efficacy in the striatum [5] and has been causally linked to learning [6]. This and other biological evidence have led to a family of mechanistic theories of learning within the basal ganglia network [7–9]. According to these models, positive and negative outcomes of actions are encoded separately in the direct and indirect pathways of the basal ganglia.

Crucially, the balance between those pathways is also controlled by dopamine [10]: An increased dopamine level promotes the direct pathway, whereas low levels of dopamine promote the indirect pathway. The above-mentioned family of basal ganglia models includes these modulatory mechanisms too. This makes the models consistent with some well-studied phenomena whereby dopamine modulates how uncertainty and risk affect decision making. For example, dopaminergic medication can bias human decision making towards or away from risk [11–14]. Further, phasic responses in dopaminergic brain areas predict people's moment-to-moment risk-preferences [15].

In summary, ample evidence suggests that dopamine bursts are related to distinct behavioral phenomena—learning and risk-taking—by way of 1) acting as reward prediction errors, affecting synaptic weights during reinforcement learning, and 2) inducing risk-seeking behavior directly. There is no obvious a priori reason for those functions to be bundled together; in fact, one would perhaps expect them to work independently, and their conflation might lead to interactions, unless some separation mechanism exists. There have been different suggestions for such separation mechanisms: it has been proposed that the tonic level of dopamine might modulate behavior directly, while phasic dopamine bursts provide the prediction errors necessary for reward learning [16]. Alternatively, cholinergic interneurons might flag dopamine activity that is to be interpreted as prediction errors by striatal neurons [17]. However, it has also been suggested that the prediction errors encoded by dopaminergic neurons might drive both learning and decision making simultaneously [18].

Curiously, even though the multi-functionality of dopamine has been noted and separation mechanisms have been proposed, interference between the different functions has, to our knowledge, never been tested experimentally. Here, we investigate this: if dopamine indeed provides prediction errors and modulates risk preferences at the same time, do these two processes interfere with each other, or are they cleanly separated by some mechanism? In other words, we test whether prediction errors are associated with risk-seeking.

A common method to provoke prediction-error related dopamine bursts in humans is to present cues and outcomes in sequential decision-making tasks, hence causing prediction errors both when options are presented, and at the time of outcome [19–21]. To test whether such prediction errors induce risk-seeking, we used a learning task in which prediction errors are followed by choices between options with different levels of risk. If there was a clear separation of roles, then risk preferences should be independent of prediction errors. Incomplete

separation, in contrast, should result in a correlation between risk preferences and preceding prediction errors. In particular, we hypothesized that positive prediction errors, occurring when expectations are exceeded, should induce risk-seeking, while negative prediction errors should lead to risk-aversion.

Overall, we found effects that were consistent with our predictions: Risk-seeking was higher when choices followed positive prediction errors than when they followed negative prediction errors. These preferences emerged gradually over the course of learning and could not be explained by any of several other known mechanisms.

## Results

### Task & theory

In this first section, we introduce our task and provide a detailed theoretical analysis of the behavior we expect from our participants. This analysis is based on models of the basal ganglia network and allows us to derive concrete predictions and models for behavior which we use for data analysis below.

**Task.** Our task consisted of sequences of two-alternative forced choice trials. On each trial, after an inter-trial interval (ITI) of 1 s, two stimuli (fractal images, Fig 1A) were drawn from a set of four stimuli and shown to the participant, who had to choose one. Following the choice, after a short delay of 0.8 s a numerical reward between 1 and 99 was displayed under the chosen stimulus for 1.5 s. Then, the next trial began. Participants were instructed to try to maximize the total number of reward points throughout the experiment.

The reward on each trial depended on the participant's choice: each stimulus was associated with a specific reward distribution from which rewards were sampled. The four reward distributions associated with the four stimuli were approximately Gaussian and followed a two-by-two design: the mean of the Gaussian could be either high or low (60 or 40), and the standard deviation could be either large or small (20 or 5), resulting in four reward distributions in total (risky-high, risky-low, safe-high and safe-low, Fig 1B). The names derive from the idea that it is "risky" to pick a stimulus associated with a broad reward distribution, since outcomes might deviate a lot from the expected outcome. Correspondingly, it is "safe" to pick a stimulus with a narrow distribution, since the outcomes will mostly be as expected.

We organize trials into three conditions: 1) "**different**": trials in which the shown stimuli have different average rewards (for example risky-high and safe-low, which have average rewards of 60 and 40 respectively), 2) "**both-high**": trials in which both stimuli have a high average reward (risky-high and safe-high, both have an average reward of 60 points) and 3) "**both-low**": trials in which both stimuli have a low average reward (risky-low and safe-low, both have an average reward of 40 points).

The stimuli shown on any given trial were selected pseudo-randomly, such that all ordered stimulus combinations (12 combinations) would occur equally often (10 times each) during each block.

**Theoretical analysis of learning and decision making.** In this section, we sketch a mechanistic theory of learning and decision-making in our task. This theory is used to derive the computational model we use in our modelling analysis (see Results/Modelling). We also use it to derive behavioral predictions (see Results/Task & Theory/Behavioral Predictions).

Our theory is based on [8,9]. Its premise is that choices are governed by competing action channels in the basal ganglia network, an assumption common to many models of the basal ganglia [22,23]. We assume that for each option $i$ in our task, there is one such action channel, and that the probability of choosing option $i$ depends on the total activation $A_i$ of that action channel. There are two contributions to this activation: excitation of magnitude $G_i$ through the

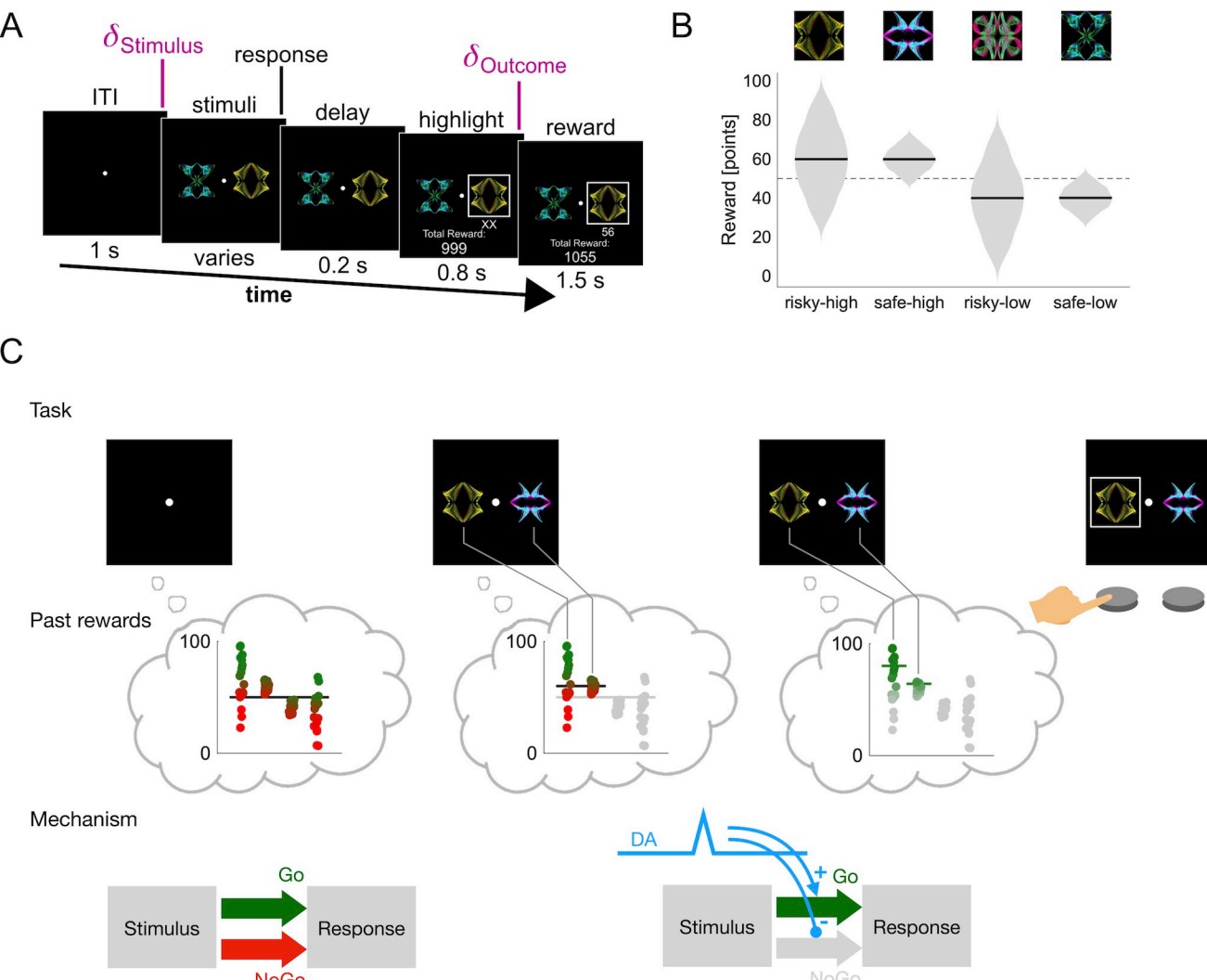

**Fig 1. A mechanistic theory of basal ganglia learning predicts prediction-error induced risk-seeking.** A) Task structure. Each trial begins with a blank screen. After 1 s, two out of four possible stimuli are selected pseudo-randomly and shown to the participant. The participant chooses one of them, which is then highlighted after a delay of 0.2 s. After another delay of 0.8 s, a point reward is displayed underneath the chosen stimulus. During each trial two separate prediction errors occur: At stimulus presentation the participant discovers whether the two displayed stimuli are better (positive prediction error) or worse (negative prediction error) than the expected average stimulus. At outcome presentation the participant discovers whether the obtained reward is higher or lower than the expected average reward of the selected stimulus. B) Reward distributions. Each stimulus (top) is associated with a different reward distribution (bottom). The distributions differ in mean (60 points versus 40 points) and standard deviation (20 points versus 5 points). Reward distributions were unknown to participants. A new set of images was used for every block. The dashed line in the background indicates the middle of the reward range, which was at 50 points. C) Events during the first half of a trial (ITI to response) according to our theory. As the blank screen appears, the participant's reward expectation (black horizontal line in the first thought-bubble) is based on all past rewards (distributions in the first thought-bubble). Above-average past rewards are encoded in the Go pathway (green dots) and below-average past rewards in the NoGo pathway (red dots). They are weighed equally since the Go and the NoGo pathways are in balance (first diagram in bottom row). As the stimuli appear, only the relevant past rewards are considered (irrelevant rewards greyed out in the middle thought-bubble), and the reward expectation changes (black horizontal line in the middle thought-bubble is higher than the black line in the first thought-bubble). The upward change in reward expectation constitutes a temporal difference prediction error which is signaled by increased dopamine transmission in the striatum (blue transient in bottom row; delta stimulus in panel A). This suppresses the NoGo pathway and enhances the Go pathway (second diagram in bottom row). As a result, below-average past rewards are ignored, and the focus is on above average past rewards (below average rewards greyed out in last thought-bubble). The stimulus with the larger spread is now valued higher (green lines in last thought-bubble) and is therefore chosen.

direct pathway (also called Go pathway), and inhibition of magnitude $N_i$ through the indirect pathway (also called No-Go pathway). These contributions are differentially modulated by dopamine (DA), mediated through the D1 and D2 receptors expressed in the direct and

indirect pathway respectively. Let $D \in [0,1]$ represent the level of dopaminergic modulation ($D = 0.5$ is the baseline level, which corresponds to a balance between direct and indirect pathway. $D = 1$ means that modulation is such that only the direct pathway contributes to action selection, while $D = 0$ means that only the indirect pathway contributes.). Then, the activation of an action channel corresponding to option $i$ can be approximated as

$$A_i = D \times G_i - (1 - D) \times N_i. \qquad \text{Eq 1}$$

This formula is motivated by (9), Eq 25, where a detailed discussion of biological plausibility is provided. What determines $G_i$ and $N_i$ for each option? According to [8,9], the direct and indirect pathway are subject to DA-dependent plasticity. Hence, $G$ and $N$ change through reward-driven learning—roughly, $G$ tracks the upper end of the reward distribution, while $N$ tracks the lower end. More accurately, we may assume that $\frac{1}{2}(G - N)$ tracks $Q$, the mean of the reward distribution, according to the rule

$$\Delta Q_i = \alpha_Q (r - Q_i), \qquad \text{Eq 2}$$

and that $\frac{1}{2}(G + N)$ tracks $S$, the spread of the reward, according to the rule

$$\Delta S_i = \alpha_S (|r - Q_i| - S_i) \qquad \text{Eq 3}$$

where $\alpha_Q > \alpha_S$ are both learning rates. Using this notation, Eq 1 reads

$$A_i = Q_i + (2D - 1) \times S_i. \qquad \text{Eq 4}$$

The impact of reward spread on action activation is *gated* by the level of DA.

As mentioned above, $D$ must take values between 0 and 1 for Eq 1 to be biologically plausible. To ensure this, we introduce a reparameterization: let $\delta \in (-\infty, +\infty)$ quantify dopamine release relative to the baseline, i.e., $\delta = 0$ corresponds to steady state dopamine release, $\delta < 0$ means that dopamine release is suppressed, and $\delta > 0$ means that dopamine release is enhanced. We can then write

$$D = \sigma(\tilde{\omega}\delta) \qquad \text{Eq 5}$$

with $\sigma(x) = (1 + e^{-x})^{-1}$ the sigmoid function and $\tilde{\omega}$ a proportionality constant. Using this notation, we can be sure that Eq 1 will produce biologically plausible results for any value $\delta$ may assume. Inserting the new parametrization into Eq 4 yields

$$A_i = Q_i + \tanh(\omega\delta) \times S_i, \qquad \text{Eq 6}$$

with $\omega = \tilde{\omega}/2$ a rescaled proportionality constant. This is the equation that we will use in our models. As we have shown, it is derived from a model of the basal ganglia pathways and parameterized such that it will always operate in a biologically plausible range.

In our task, the $Q$ and $S$ of the four action channels should converge to the means and spreads of the four reward distributions. With DA at baseline (i.e., $D = 0.5$ or $\delta = 0$) activation is proportional to $Q$, and hence to the mean reward. Choices should thus be biased towards options with high mean rewards. If DA levels are increased (i.e., $\delta > 0$), the learned spread $S$ contributes positively to action activation, biasing choices towards risky options. If, on the other hand, DA levels are below baseline (i.e., $\delta < 0$), the learned spread $S$ reduces action activation, biasing choices towards safe options.

**Theoretical analysis of reward prediction errors.** In this section, we focus on how the participant's reward prediction should theoretically change over the course of a trial. This is based on the theory of temporal difference learning [24], which has been applied to describe dopaminergic responses to rewards (2). The basic assumption of these theories is that

participants maintain a prediction of upcoming rewards at all times. This prediction is based on learned estimates $Q$ of average rewards associated with the four stimuli. At the beginning of a trial (before the stimuli appear), participants do not have any specific information to base their prediction on. Given that our task design contains a fixed ITI and trials have the same structure throughout, we may assume that participants anticipate the appearance of stimuli at a certain time after the initial fixation. The corresponding reward prediction at that time should then be an average over all possibilities, i.e., an average over the learned values of all options that might occur.

After the options appear, participants should adjust their reward prediction based on the learned values of the displayed options. We take the participants' updated prediction to be the average learned value over the presented options. The updated prediction should differ across conditions: if the participants learned accurate estimates of the values, their prediction would be 60 points in the both-high condition and 40 points in the both-low condition.

This change in participants' reward expectation through the appearance of reward-predicting stimuli constitutes a prediction error called **stimulus prediction error** $\delta_{stimulus}$, previously described in [25], and should cause phasic DA activity (2). The magnitude of that prediction error is given by

$$\delta_{stimulus} = mean_{shown}(Q) - mean_{all}(Q). \qquad \text{Eq 7}$$

If the values of the stimuli are learned with reasonable accuracy, this prediction error should be 60−50 = 10 in the both-high condition and 40−50 = −10 in the both-low condition. After successful learning, we hence expect a positive stimulus prediction error in the both-high condition and a negative stimulus prediction error in the both-low condition.

It is important to note that the stimulus prediction error is a reward prediction error that occurs at the time of stimulus onset, not an error in stimulus prediction. The identity of the stimulus is relevant only insofar as it is related to reward expectations.

Next, participants will make a choice. Now, their reward expectation is the value of the option they chose. Finally, a reward is displayed, forcing participants to update their reward estimate again. This second prediction error—the difference between the learned value of the option and the actual reward received—we call the **outcome prediction error** $\delta_{out}$. Its magnitude is given by

$$\delta_{out} = r - Q_{selected}. \qquad \text{Eq 8}$$

It is the outcome prediction error that drives learning about the stimuli (see Eqs 2 and 3 above). Again, note that the outcome prediction error is a reward prediction error at the time of outcome presentation. In our parlance, stimulus prediction errors and outcome prediction errors exist within the same framework—they are both reward prediction errors but happen at different times.

In summary, our analysis reveals that two prediction errors (and two corresponding DA responses) should occur in each trial: first the stimulus prediction error shortly after the presentation of the options, and second the outcome prediction error after the presentation of the reward.

Leveraging latent variables extracted from trial-by-trial modelling (see section Modelling below) and measurements of our participants' pupil diameter, we found physiological evidence for surprise linked to the two prediction errors we described in this section (see Fig A in S1 Text). While this does not directly confirm the occurrence of the dopamine responses we describe here, it does suggest that both stimulus onset and outcome presentation trigger cognitive processes related to reward expectation.

Next, we will combine this analysis of prediction errors with the theory of learning and decision making in the basal ganglia that we discussed above, hence arriving at the Prediction Errors Induce Risk Seeking (PEIRS) model.

**The PEIRS model.** The key novel idea of PEIRS is to connect two dopamine-related phenomena—the dopaminergic modulation of risk preferences on the one hand, and the dopamine responses to changes in reward expectation on the other.

The model is based on the learning rules Eqs 2 and 3 from above:

$$\Delta Q_i = \alpha_Q (r - Q_i),$$

Eq 9

$$\Delta S_i = \alpha_S (|r - Q_i| - S_i)$$

Eq 10

We add a choice rule based on the activation $A_i$ of action channels as modelled in Eq 6. Those activations are turned into choice probabilities using a conventional softmax choice model [26]. Crucially, we propose that the relevant modulatory dopamine release ($\delta$ in Eq 6) is the release that corresponds to the stimulus prediction error in Eq 7, i.e.

$$\delta = \delta_{stimulus}.$$

Eq 11

In other words, we propose that the stimulus prediction error (a reward prediction error that occurs at the time of stimulus presentation) might cause dopamine release or inhibition, which will then modulate the risk preferences in the choice between the stimuli through modulation of the basal ganglia pathways.

Using Eq 11 and Eq 6, we finally arrive at

$$A_i = Q_i + \tanh(\omega \delta_{stimulus}) \times S_i,$$

Eq 12

for the activation of the action channels, which are turned into choice probabilities via a regular softmax rule, $p_i = \frac{exp(\beta A_i)}{\sum_j exp(\beta A_j)}$. This model, completely defined by Eqs 9, 10 and 12, encodes our hypothesis: that prediction errors might affect risk preferences by modulating the balance of the basal ganglia pathways.

**Behavioral predictions.** In addition to developing a mathematical model of learning and decision making, we can also use our theory to derive task-specific behavioral predictions from our theory (see Fig 1C for a schematic representation). We have seen that in the matched-mean conditions (both-high and low), the presentation of the options should cause a prediction error (positive and negative, respectively), and hence a transient change in DA levels (increase and decrease, respectively) in the striatum during the choice period (Fig 1C, mechanistic level). We have also seen that DA levels affect choices through modulation of the basal ganglia pathways: increased DA makes people risk-seeking, decreased DA makes them risk-averse (see Eq 6). If the average reward is similar for two options (as it is in the both-high and the both-low condition), these risk preferences should be the decisive factor in decisions (Fig 1C, task level and past rewards level).

Taken together, these premises suggest that the risk preferences in the both-high condition should be different to those in the both-low condition—risk-seeking should be stronger in both-high than in both-low, because the stimulus prediction error is higher in both-high than in both-low. This should be the case even if other, condition-independent risk preferences (such as a general risk-aversion) occur in addition to the effect we propose.

If there were no other substantial risk effects, we could make another even more specific prediction: that we should see a preference for the risky stimulus in the both-high condition (depicted in Fig 1C), and a preference for the safe stimulus in the both-low condition. These

effects should appear gradually, since they require that both mean and spread of the reward distributions are learned. Specifically, risk preferences in the both-high and both-low conditions should appear slower than value preferences in the difference condition. This follows from the underlying plasticity rules: one can show that the learning rate for spread must always be lower than the learning rate for value [9]. In addition to this, a reasonably accurate value estimate is required for the spread estimate to converge; this also contributes to a higher learning speed for value compared to spread.

In summary, we have derived two predictions: 1) We should see a difference in risk preference between conditions, and 2) we should see gradually emerging risk-seeking in the both-high and risk-aversion in the both-low condition. To arrive at these predictions, we took into account neural mechanisms. However, the predictions are purely on the level of behavior. In the empirical part of this study, we focus on behavior, conscious that this will only provide indirect evidence for the neural underpinnings of the proposed mechanism. However, it might reveal a previously unknown effect with a clear, plausible biological explanation. We provide further predictions based on neural signals in the Discussion.

## Behavior

In the previous section, we introduced a novel reinforcement learning task and performed a theoretical analysis to derive behavioral predictions. In this section, we present the results of testing these predictions.

We recruited a cohort of participants (N = 30, 3 excluded, see Fig B in S1 Text) and recorded their behavior in the task described above. Each participant performed four blocks of 120 trials. During each block, all six possible stimuli pairings occurred equally often. Each block used a new set of four stimuli, mapped to the same four distributions. Participants made their choices on average 0.97 s (standard deviation 0.51 s) after stimulus onset.

First, we investigated whether participants' performance improved during the task. Based on the premise of associative learning, we expected choice accuracy (i.e., likelihood of choosing the option with the higher average reward) to increase gradually over trials. To confirm this, we focused on choices in the difference condition, where participants had to choose between stimuli with different average rewards. We found that indeed, the probability of choosing the stimulus with the higher average reward increased gradually over trials across the population. Average performance differed from chance level with high significance (Fig 2A, t-test, t(27) = 31.9, p < 0.001) and approached its asymptote in the second half of the block (see Fig C in S1 Text for choice proportions in the different condition split by stimuli combinations). These findings suggest that our participants successfully used associative learning in our task, confirming a basic assumption of our theory.

Next, we tested our first prediction—that there would be a difference in risk-preferences (i.e., in the likelihood to choose the risky option over the safe option) between the both-high and the both-low condition. Specifically, we predicted higher risk-seeking in both-high than in both-low. For each condition, we investigated the likelihood of choosing the stimulus with the broad distribution (risky) over the stimulus with the narrow distributing (safe). Preferring risky over safe was considered risk-seeking, preferring safe over risky was considered risk-averse. We computed the average difference in risk-preference between conditions for each participant across all trials. We found that most of the participants were more risk-seeking in the both-high condition than in the both-low condition (Fig 2B; two-tailed paired t-test: t(27) = 3.58, p = 0.0016), which confirmed our first prediction.

We then tested our second prediction—that participants would be risk-seeking in the both-high condition and risk-averse in the both-low condition, and that these effects would emerge

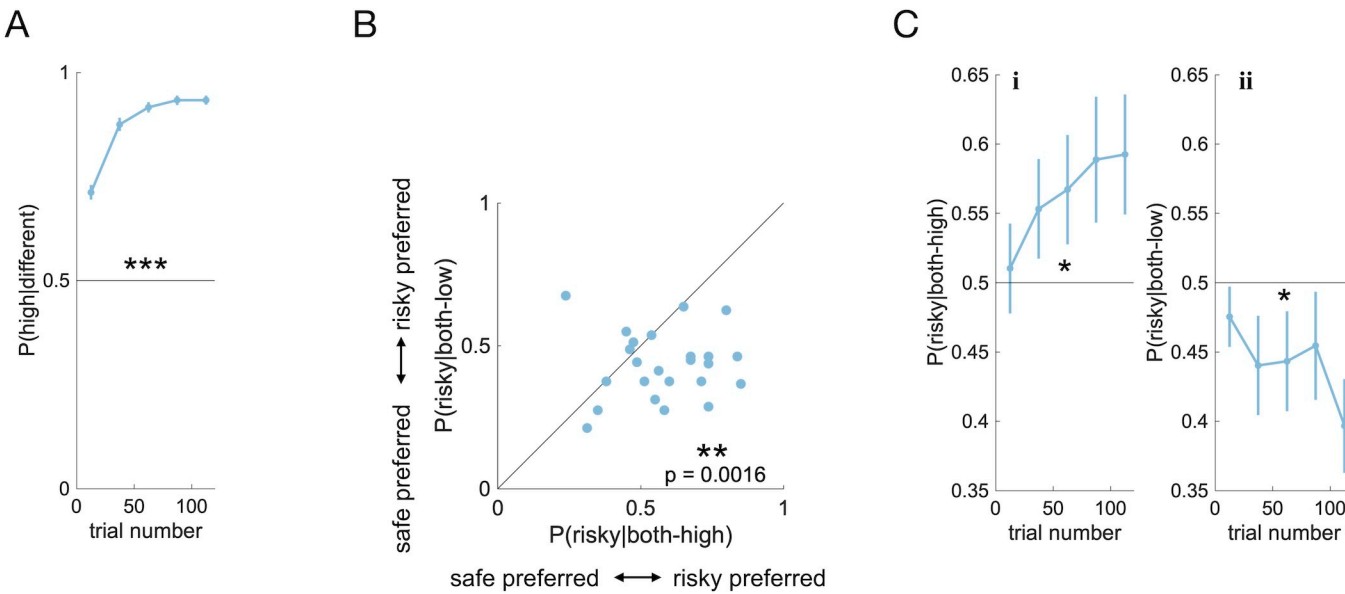

**Fig 2. Condition-specific risk preferences emerge gradually.** A) Probability of choosing a stimulus with high average reward over a stimulus with low average reward, as a function of trial number. Choices were binned according to trial number. For each bin, we show the mean (dot) and SE (bar) over subject means. The stars (*: p < 0.05, **: p < 0.01, ***: p < 0.001) indicate that the population mean across all trials is significantly different from indifference (0.5). B) Correlation between risk preferences in the both-high mean and the both-low condition across all trials. Each point represents one participant. Preference for the risky stimulus if mean rewards are high is plotted against preference for the risky stimulus if mean rewards are low. If a point falls below the diagonal, the participant was more risk-seeking for high-mean stimuli than for low-mean stimuli. The stars indicate that the population mean is significantly below the diagonal. C) Probability of choosing the risky stimulus over the safe stimulus, i) if mean rewards are both high, ii) if mean rewards are both low, as a function of trial number. Data represented as in A. Data and code to reproduce this figure can be found at https://doi.org/10.5281/zenodo.5111690.

gradually and more slowly than increases in task performance. As predicted, we found significant risk-seeking in the both-high condition (Fig 2Ci; two-tailed t-test: p = 0.0343, t(26) = 2.23) and significant risk-aversion in the both-low condition (Fig 2Cii; two-tailed t-test: p = 0.0317, t(26) = -2.27). These preferences emerged gradually as a function of trial number, at a slower rate than the preferences for the high over low value stimuli (compare Fig 2A against Fig 2Ci and Fig 2Cii; mixed effects model (see Methods): p = 0.001). The data thus confirmed our second prediction in all aspects.

In summary, in data recorded from the task described above, we found the two behavioral effects we predicted. Our findings provide initial evidence for a behavioral link between prediction errors and risk preferences. In the next section, we use computational modelling to compare our explanation of the measured effects to alternative explanations.

## Modelling

In the previous section, we have shown that effects like those predicted by our theory can be found in experimental data. Our analysis so far rested on the assumption that participants know the ground-truth means and standard deviations of the reward distributions. However, these statistics have to be learned when participants perform the task. To capture this learning process, trial-by-trial learning models can be fit to the data. These models allow us to answer some important questions that have not yet been addressed:

1. Are there alternative explanations for the observed effects?

2. Does our theory fit the data better than existing theories?

In this section, we use computational modelling to answer these questions. To test our theory against alternative explanations, we use simulations as well as model comparison techniques [27].

**Models.** Associative learning in tasks like ours is commonly described with the Rescorla-Wagner (RW) model [1]. All models we use here are variants of this base model. RW is also the first type of explanation for the effects we observed—it has been shown that even basic associative learning can yield risk preferences through sampling biases [21].

The second type of explanation involves the utility of reward points: risk-aversion as well as risk-seeking have been explained as consequences of nonlinear utility functions [28]. We consider the s-shaped family of utility functions as particularly relevant, as it encodes important aspects of prospect theory: a reference point relative to which outcomes are evaluated, and different signs for the curvature of the utility function on both sides of the reference point. The corresponding model is called s-shaped UTIL.

The third type of explanation is based on RW with variable learning rates. It has been observed that humans use different learning rates for positive and negative outcomes [29], and that this can lead to risk preferences [21,30]. We implement this in the model pos-neg RATES.

Finally, the explanation that we propose in this study is that prediction errors induce transient risk preferences, as detailed above. We represent our hypothesis in the PEIRS model, which we have introduced above.

The models described here encode what we consider to be the most relevant alternative explanations of our effect: they are all well-grounded in empirical research and have all been related to risk preferences before. We will therefore provide the corresponding results in the main text, including simulations and pairwise model comparisons (see below). In addition to those, we tested a wide array of other, less prominent models. All models we investigated are described in detail in the methods section, and detailed results are provided in S1 Text. However, in the main text we only include them in the overall model comparison (Fig 3C) and introduce them via a short summary, as follows.

Besides s-shaped utility, we tested several other types of utility functions: a concave utility function (concave UTIL), a convex utility function (convex UTIL) and an inverse s-shaped utility function (inverse s-shaped UTIL). Together with s-shaped UTIL, those families of functions cover all basic curvature types, among them those that are popular in neuroeconomics (the concave type of expected utility theory and the s-shaped type of prospect theory), as well as some more exotic types (convex and inverse s-shaped).

Learning rates might depend on the valence of the prediction error, as codified in the pos-neg RATES model. However, they might also be modulated by other variables. We included one model in which learning rates are allowed to be modulated by the reward variability (the variance RATES model, which features different learning rates for the high-variance and the low-variance stimuli). We also included a model that aims to capture attention effects (the attention RATES model): here, learning rates depend on the surprise (the absolute prediction error) experienced on the current trial. The core idea is that very surprising outcomes might draw relatively more attention and might hence be committed to memory more thoroughly.

Not only learning rates can be modulated—prediction errors, too, could be subject to context-specific adaptation (in some cases, such as scaling, modulation of the learning rate is mathematically equivalent to modulation of the prediction error). In particular, there is substantial evidence that prediction errors might adapt to the statistics of the reward distribution [31–33]. We included this hypothesis as a potential explanation through the scaled PE model.

Finally, we included two variations of our explanation (PEIRS): first, we considered the possibility that predictions, not prediction errors might modulate risk-seeking (the PIRS model, Predictions Induce Risk Seeking). Second, we included a model in which risk preferences are

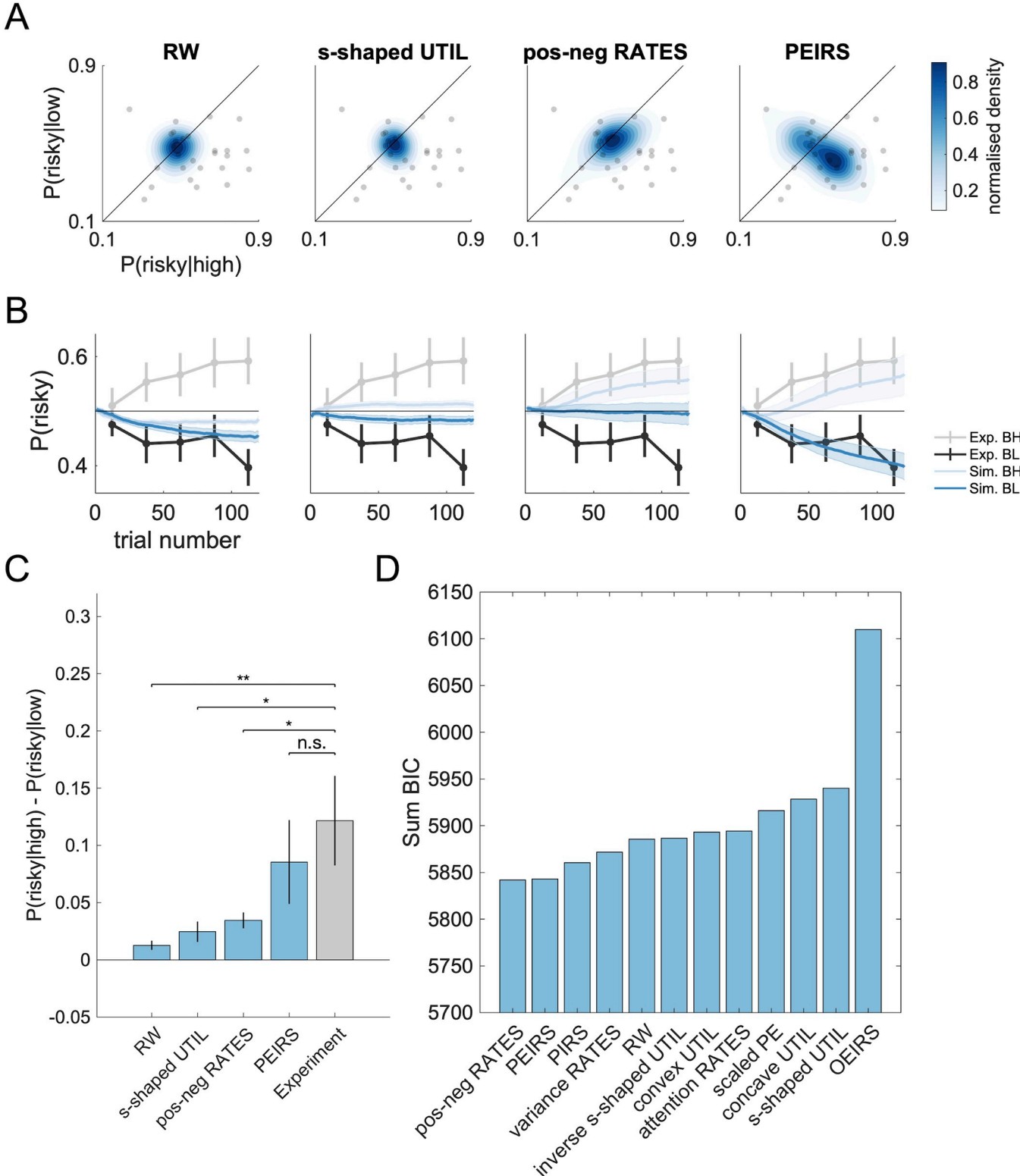

**Fig 3. PEIRS either fits better or reproduces risk preferences better than all alternative models.** A) Population risk preferences in simulated datasets. The risk preference distributions extracted from simulated data are plotted as blue contour plots. The corresponding experimental data is superimposed as a grey scatterplot (identical in all panels). The shading corresponds to estimated probability density functions and is scaled to match each distribution's density value range for better visibility. B) The temporal evolution of risk-preferences in simulated datasets. Risk preferences were extracted from simulated (Sim.) datasets, split between conditions (BH: Both-High, BL: Both-Low) and plotted as a function of trial number (light blue for the both-high condition, dark blue for the

both-low condition). The shaded areas indicate the SE across simulated participants. The simulated data were smoothed using a moving average. The corresponding experimental (Exp.) data is superimposed in grey (light grey for the both-high condition, dark grey for the both-low condition), represented in the same way as in Fig 2C. C) Difference in risk preferences between conditions, for simulated and empirical datasets. Simulated datasets are shown as blue bars, the experimental dataset is shown as a grey bar. We show the mean (bar) and the SE (errorbar) of participant means (averaged over experiment repetitions for simulated datasets). D) Overall model comparison results. We show the BIC summed over all participants, for all models that were tested. Lower BIC indicates a better fit. Data and code to reproduce this figure can be found at https://doi.org/10.5281/zenodo.5111690.

modulated by the outcome predication error on the previous trial (the OEIRS model, Outcome Errors Induce Risk Seeking).

**Simulations.** First, we wanted to know which of the models could reproduce (and hence explain) the observed effects. To check this, we first fitted all models to our dataset and extracted maximum-likelihood parameter estimates for each participant. We then used these parameters to simulate behavior in our task with all candidate models to generate synthetic datasets. Finally, we analyzed these synthetic datasets in the same way as the experimental dataset. In our analysis, we considered difference in risk preferences between conditions as well as the distribution of risk preferences across the population (Fig 2B) and the evolution of risk preferences over trials (Fig 2C).

We found that our hypothesis reproduced the effect best: the PEIRS model captures the overall distribution (Fig 3A) and the condition-specific gradual emergence (Fig 3B) of risk preferences across the population better than RW, s-shaped UTIL and pos-neg RATES (see Fig D in S1 Text for likelihood ratios). It further and produces the most realistic effect size among these models (Fig 3C, empirical effect size: 0.122, simulated effect size with PEIRS model: 0.086, no significant difference between experimental data and simulated data, two-tailed paired t-test, t(26) = -1.84, p = 0.0767).

The main alternative models (RW, s-shaped UTIL, pos-neg RATES) predicted very narrow distributions for the risk preferences, which is at odds with our empirical results (Fig 3A). They also fail to reproduce the emergence of differential risk preferences over trials (Fig 3B): preferences comparable to those observed in the experiment either emerge only in one condition (see pos-neg RATES), or not at all (RW, s-shaped UTIL).

With respect to our main effect—the difference of risk preferences between conditions—all alternative models predicted effect sizes that differed significantly from the empirical findings (Fig 3C, two-tailed paired t-tests, s-shaped UTIL: t(26) = -2.19, p = 0.038, RW: t(26) = -2.78, p = 0.010, p = 0.013, pos-neg RATES: t(26) = -2.4, p = 0.025).

We conclude that among RW, s-shaped UTIL and pos-neg RATES, our theory, embodied by the PEIRS model, was the best explanation for the observed risk preferences (with respect to distribution, emergence and differences between conditions). All alternative explanations were falsified by the simulations. The same holds true for all other models we tested (see Fig E in S1 Text), with the single exception of inverse s-shaped UTIL, which captures both the emergence of differential risk preferences and the difference between conditions (but is inferior to PEIRS in capturing the distribution of risk preferences, see Fig D in S1 Text).

**Model fit.** After showing that the PEIRS model can reproduce the risk preferences that we measured, and falsifying most alternative explanations, we turned to use model comparison techniques to test more formally whether our dataset provides evidence for or against the PEIRS model. A model comparison can reveal which of several explanations that describe the effect in question equally well is the most appropriate.

We conducted a standard model comparison based on the Bayesian Information Criterion (BIC), which is a classical metric to assess how parsimoniously a model describes a set of data [34]. Our analysis is carried out on the population level. This means under the assumption that the entire population uses a single one of the candidate models (albeit with individual sets of

parameters), we determine which model is most likely to be the one that is used. We validated this method by performing a model recovery analysis (Fig F in S1 Text). That analysis suggests that for datasets like ours there is a slight risk of false negatives (i.e. attributing the dataset to RW when it was really generated from a more complicated model). The risk of false positives, on the other hand, is very low, especially for the PEIRS model.

The results of the model comparison suggest that the best description of our population is afforded by the pos-neg RATES model, very closely followed by the PEIRS model (see Fig 3D). The other models are worse to various degrees; the OEIRS model is the worst by a substantial margin.

In relation to the typical differences in BIC between our models, the difference between pos-neg RATES and PEIRS seems negligible—we may think of them as about equally parsimonious with respect to our dataset. The fact that PEIRS has more free parameters further suggests that it captures more variance than pos-neg RATES, given that the number of parameters is traded against the goodness of fit in the BIC metric. This is in line with new results that replicate our findings: [35] show that PEIRS describes their population better than RW, despite its increased complexity.

Further, we can see a very clear difference in BIC between PEIRS and the inverse s-shaped UTIL model, which was the only model not falsified by the simulations. This result is highly non-trivial, given that the PEIRS model has two free parameters more than the s-shaped UTIL model, and hence receives a stronger penalty in the BIC metric.

**Conclusions of modelling.** Taking together the results of the simulations and the model comparison, a clear picture emerges: all models but PEIRS and inverse s-shaped UTIL were falsified in simulations, while the model comparison suggests that PEIRS and pos-neg RATES describe our data best. This implies that the effects of interest (risk preferences) are best explained by the PEIRS hypothesis, while the pos-neg RATES model seems to capture another feature of our dataset well.

## Discussion

Different behavioral phenomena—learning from prediction errors and biased risk-preferences–are attributed to the same neurotransmitter, dopamine. The common neural substrate led us to hypothesize that there might be an association between prediction errors and risk preferences on the level of behavior. To investigate this, we first used a theory of basal ganglia learning to develop a model that represents the mechanism that we propose, and to derive concrete predictions of the hypothesis. Second, we tested our predictions using a task where reward prediction errors are immediately followed by decisions that involve risk. We found that reward prediction errors and the probability of subsequent risk-taking are positively correlated: positive reward prediction errors induce risk-seeking, negative ones inhibit risk-seeking. Finally, we used trial-by-trial modelling to test the mechanism that we propose, alongside various alternative explanations. A combination of simulations and model selection techniques revealed that out of all tested models, our hypothesis is the most likely explanation for the effects that we observed.

Our results are consistent with our initial hypothesis: the two roles of dopamine (teaching signal and risk-modulator) interfere with each other. This study hence provides evidence *against* the conjecture that the roles of dopamine are well separated. Our conclusion fits in well with other recent findings: recent research has shown that phasic dopamine correlates with motivational variables [36] and movement vigor [37] just as well as with reward prediction errors. Together, these findings cast doubt on the separation into tonic and phasic and on separations in general.

### Predictions or prediction errors?

Our theory rests on the occurrence of stimulus prediction errors to explain risk preferences. Generally speaking, positive stimulus prediction errors would explain risk-seeking, while negative prediction errors would explain risk-aversion. We indeed observe risk-seeking as well as risk-aversion (see Fig 2C) and find that those effects are best explained by the PEIRS model, which features positive as well as negative prediction errors (see Fig 3). While this explanation is consistent in itself and compatible with the data we collected, one might question its biological plausibility, on the grounds that reward predicting stimuli are known to elicit dopamine bursts, but not dips. For example, a classical study shows increased dopamine activity as a response to reward-predicting stimuli, even for stimuli that predict a relatively small reward [33].

There seems to be a contradiction between these results and our assumptions, but they are in fact compatible. To see this, one must consider the details of the trial structure in the study in question: while our study had a fixed ITI, many classic studies (such as [33]) have a variable, random ITI. This means that while our participants could predict the time of stimulus onset perfectly, the onset was surprising for the subjects in [33]. This is a crucial difference: if the time of stimulus presentation can be predicted, a reward expectation for that moment can be formed, and the value of the stimuli can be compared to this expectation. If the time of stimulus onset is unknown, then any stimulus that appears will first and foremost be compared to the possibility that nothing happens at that moment. The result of that comparison must be positive, as the occurrence of reward predictors is always better than their absence.

This difference is manifest in neural recordings. For example, fixed ITIs were used in combination with stimuli that predicted different reward sizes in [38]. The dopamine activity shown in Fig 2E and 2F of that study clearly dips below the baseline level for stimuli that predict relatively small rewards. This suggests that the assumptions of the PEIRS model are biologically plausible for tasks with fixed ITIs, such as the one we are using in this study.

We did nevertheless include a model based on reward predictions, not prediction errors, which assumes dopamine responses like those in [33]. That was the PIRS (Predictions Induce Risk Seeking) model, which we included in our model comparison (Fig 3D). Simulations are shown in Fig E in S1 Text. Both simulations and model comparison show clearly that PIRS does not explain the data well.

In summary, we may conclude that prediction errors, but not predictions, might induce risk-seeking in our task, and that the underlying mechanisms are consistent with what is known about dopamine release in tasks with predictable timing.

Another question that might arise in this context is about the role of the outcome prediction error of the previous trial. According to our theory, that prediction error should be broadcasted by the dopamine system just like the stimulus prediction error and might therefore also affect risk preferences. Of course, there is a difference in timing: on average, the choice follows the stimulus prediction after 0.97 s, while the delay between outcome prediction error and choice on the next trial is on average 3.47 s. We might thus expect that the impact of the outcome prediction error, if at all observable, might be much weaker than that of the stimulus prediction error. A supplementary analysis similar to those displayed in Fig 3A–3C confirmed this: there is no evidence for an association of risk preferences and outcome prediction error of the previous trial in our dataset (Fig E in S1 Text).

**Relation to behavioral economics.** Decision making under uncertainty has been extensively studied in behavioral economics. One main finding in this field, codified in prospect theory, is that humans tend to be risk-averse if decisions concern gains, and risk-seeking if decisions concern losses [28]. However, those classic findings rely on explicit knowledge about

the probabilities involved in the decisions. Several more recent studies indicate that risk preferences reverse when risks and probabilities are learned from experience (i.e., by trial and error): if learning is incremental and based on feedback, humans tend to make risky decisions about gains and risk-averse decisions about losses [39]. This reversal has been termed the description-experience gap and is considered a "major challenge" for neuroeconomics [30]. In cognitive neuroscience and psychology, some studies have reproduced this phenomenon [40], while others report risk-aversion in the gain domain [21].

Our task differs somewhat from the tasks studied in the description-experience gap literature, since we only use gains (reward points are always positive). However, humans often evaluate outcomes with respect to a reference point [28]. In the context of our task, it seems plausible that rewards are evaluated relative to the average of previous rewards, or relative to the middle of the experienced reward range, which rapidly converges to 50 points within the first few trials. Rewards under 50 (and hence the majority of the rewards that result from choosing stimuli 3 and 4) would then be considered losses. Decisions between the two low-valued stimuli (both-low condition), would fall in the loss domain. Under this perspective, our results are in line with the description-experience-gap, and differ from those of [21]. The difference might be due to the degree of implicitness of the knowledge that is gained during the task: Niv et al. used classical bimodal reward distributions (e.g., 40 points with probability 50%, 0 points otherwise) which participants might be able to recognize as such after a few trials. Here, we used high-entropy reward distributions (normal distributions, see Fig 1B), which could not be mapped onto bimodal gambles, and thus made anything but implicit learning intractable.

**Relation to memory theories.** For our behavioral results, interpretations other than our dopaminergic explanation may be evoked: the behavior in a similar task [40] was interpreted as the result of memory replay: experiences ("Obtained reward X after choosing option Y") might not only be used for immediate value updates but might also be stored in a memory buffer. This buffer can then be used for offline learning from past experiences in times of inactivity, such as during the inter-trial interval. [40] proposed that experiences are more likely to enter the buffer if they are extreme. If entering the buffer is biased in this way, then so are the values learned from replaying those experiences. In our task, extreme might mean that the reward was extremely high or low. The corresponding bias would drive choice towards the stimuli that produce the highest rewards, and away from those that produce the lowest, and thereby lead to a pattern similar to the one we observed.

Even though it is not feasible to represent the memory buffer model in our framework, our inverse s-shaped UTIL model does capture the idea of overweighting of extreme experiences. The simulations of that model (Fig E in S1 Text) show that such overweighting can indeed reproduce the risk preferences that we observed, which ties in well with the results of [40]. However, our model selection procedure (Fig 3D) suggests that PEIRS still explains the data better. One big difference between the two explanations is that risk preferences can flexibly appear and disappear in PEIRS. The memory buffer theory (and equivalently the inverse s-shaped UTIL model) on the other hand attribute them to distortions in the learned values, and hence predict more persistent preferences.

Another potentially relevant phenomenon based on memory effects was reported recently: [41] show that both reward tracking itself, as well as episodic memory, are enhanced in high-risk environments. In our context, this might mean that learning is boosted for stimuli that regularly produce large prediction errors, i.e. the risky stimuli, relative to the safe stimuli, either through directly boosting the reward learning process, or through boosted memory replay as described above. We included two models that could capture such effects: the variance RATES model allowed different learning rates for risky and safe stimuli, while in the

attention RATES model, surprise could boost learning in all conditions. However, neither of the two models could explain the effects we observed, suggesting that the fascinating effects they describe are distinct from the effects that drive risk preferences in our experiment.

**Relation to utility models.**   Several of the models we have tested are based on nonlinear utility of rewards. The central idea of these models is that participants might not find an outcome of 100 points twice as rewarding as an outcome of 50 points. If the perception of reward is distorted in this way, risk preferences might arise as a consequence [28]. None of the tested utility models constituted a better explanation of our effects of interest. However, one of them (inverse s-shaped UTIL) at least reproduced the trial-by-trial emergence of risk preferences well (Fig E in S1 Text, panel C). How should this and the other utility-related results be interpreted? We see two issues with utility models.

The first issue relates to the level of explanation: utility functions can be used to capture behavioral effects, i.e., they can provide a compact description of certain aspects of behavior (such as risk-seeking). Such models might well be used to make predictions about behavior. What they cannot provide us with, however, is an explanation on the level of neural processes (this distinction was phrased as 'aggregate' versus 'mechanistic' in [27]). In fact, it might well be that for a given neural process, one may find a utility function that can compactly summarize the effects it has on behavior. These two descriptions then concern different levels of description, and comparing them might not be meaningful [27]. In this study, we derive a mechanistic model (PEIRS), and show that it is the best explanation of the effects of interest. Our level of description is thus the mechanistic one.

The second issue is generalizability and affects convex and inverse s-shaped utility functions in particular. Concave and s-shaped utilities are well documented and embedded in broader theories (expected utility theory and prospect theory, respectively). The effects that they describe appear in many different situations—they seem to capture a fairly general aspect of behavior (which perhaps relates to a general mechanism in the brain). In contrast, convex and inverse s-shaped utility functions only describe behavior in some very specific tasks (see [42] for behavior that is well described by convex utility) but fail to generalize to others. In these cases, one might be overfitting the concept of utility, at the price of specificity. At the extreme, it is conceivable that most phenomena can be explained as an effect of non-linear utility but require a specific (and perhaps quite non-trivial) utility function for each case. From the standpoint of generalizability, it seems thus seems appropriate to test established theories such as prospect theory as serious alternative explanations. Utility functions tailored to the behavioral effect in question seem more problematic.

Overall, we find that neither our empirical results nor general epistemological considerations indicate that much emphasis should be put on utility models in the context of our task and our goal (to understand the neural mechanisms that cause risk preferences).

## Further experimental predictions

In this paper, we present a theory based on neural effects, and test some of its behavioral predictions. Our predictions were confirmed in the experiment; additional evidence for our theory could be gained using a trial-by-trial modeling approach. Other, more elaborate tests on the level of behavior are possible. For example, one might increase the set of stimuli and indicate a subset of available stimuli before the trial begins. The primed subset should affect expectations and hence stimulus prediction errors. With such manipulations, one might be able to induce opposite risk preferences in the same set of options, depending on the primed subset. However, no behavioral experiment can prove that our theory is correct on the level of neural mechanisms. More direct measurements are needed to establish this.

Correlational studies are possible. They should have sufficient time resolution to differentiate the prediction errors at different times during the trial, such as electrophysiology, voltammetry or similar. With those, one could gain more direct measurements of the PEs, and hence resolve more clearly how they are associated with subsequent risk-seeking. Another possibility are causal studies. For example, optogenetic tools could be used to elicit or suppress prediction errors just before choices involving risk, for instance by stimulation VTA dopamine neurons at the time of stimulus onset.

**Conclusion.** In summary, we demonstrate that a biologically inspired theory of basal ganglia learning predicts an interaction between prediction errors and risk-seeking. This is based on dopamine's dual role in learning and action selection. We present behavioral data that matches these predictions.

## Methods

### Ethics statement

Participants were recruited from the Oxford Participant Recruitment system and gave informed written consent. All experimental procedures were approved by the Oxford Central University Research Ethics Committee, approval number R45265/RE004.

### Participants

We tested 30 participants (15 female, median age: 26, range: 18–42). Our participants did not suffer from visual, motor or cognitive impairments. Participants were given a written set of instructions, as well as an oral instruction. They were first provided with a description of the task sequence. Then, they were told that the rewards were random, but nevertheless higher on average for some shapes than for others. Finally, they were advised to get as much total reward as possible and that their compensation would be between 8 and 12 GBP, depending on their performance. After the task, all participants received a compensation of 10.50 GBP.

Our results are based on 27 of the 30 participants. Three participants were excluded from the analysis due to their failure to understand the task: we evaluated the participants' understanding of the task by scoring their preferences in different-mean choices during the second half of the blocks. Participants were included in the analysis if they chose the high-valued option in more than 65% of those trials (Fig B in S1 Text).

### Emerging preferences

To test whether value preferences emerged faster than risk preferences, we used a linear mixed effects model to model choices. As predictors, we included fixed effects of decision type, trial number, and their interaction. Here, decision type was defined as value decision (difference condition) versus risk decision (both-high and both-low conditions). We also included random effects for all predictors and a random intercept, by participant. A likelihood ratio test for the fixed effect of the interaction between trial number and decision type revealed a significant positive effect ($p < 0.001$; based on comparing the empirical likelihood to a distribution of likelihoods obtained from 1000 Monte Carlo simulations of data from the model without the fixed effect, as implemented in MATLAB's 'compare' function).

### Models

The models we used in this study are all variants of the RW model [1]. They consist of learning rules for latent variables, such as value and spread, and choice rules that convert latent

variables into choice probabilities. Below, we provide these rules for all models, along with some auxiliary rules as necessary.

For all models, $i \in \{1,2,3,4\}$ is the stimulus index, and $Q_i$ is the value of stimulus $i$. The index $j$ is used for the options on screen in each trial (for example, $j = \{1,3\}$ if stimuli 1 and 3 are shown). The initial values at the beginning of each block are denoted by $Q_0$. The update rules are only applied to the values and spreads of the chosen stimuli; the values and spreads of unchosen stimuli do not change. The probability of choosing stimulus $i$ is denoted by $p_i$, the received reward is denoted by $r$.

**RW.** In the RW model, learning is driven by differences $\delta_{outcome}$ between the reward expected and received:

$$\delta_{outcome} = r - Q_{chosen} \qquad \text{Eq 13}$$

$$\Delta Q_{chosen} = \alpha \times \delta_{outcome} \qquad \text{Eq 14}$$

Here, $\alpha$ denotes the learning rate. Learned values were linked to choice probabilities via a standard softmax rule [26]:

$$p_i = \frac{exp(\beta Q_i)}{\sum_j exp(\beta Q_j)} \qquad \text{Eq 15}$$

The RW model has free parameters $\alpha \in [0,1]$, $\beta > 0$ and fixed parameters $Q_0 = 50$.

**Concave utility.** To allow for concave subjective utility of reward points in our experiment, we used an exponential family of functions [43], which we adapted to our reward range through appropriate scaling ($\sigma$) and shifting ($m$):

$$z = \frac{r - m}{\sigma} \qquad \text{Eq 16}$$

$$U = m + \sigma \times \frac{1 - exp(-k_{concave}z)}{k_{concave}} \qquad \text{Eq 17}$$

The nonlinear utility of reward enters through the computation of the prediction error:

$$\delta_{outcome} = U - Q_{chosen} \qquad \text{Eq 18}$$

Updates are computed with Eq 14, choices are modeled using the softmax rule in Eq 15. The concave UTIL model has free parameters $\alpha \in [0,1]$, $\beta > 0$, $k_{concave} > 0$. and fixed parameters $Q_0 = 50$, $\sigma = 50$, $m = 50$.

**Convex utility.** For completeness, we also include a model that allows for convex utility functions. Although those are not as often mentioned, there is non-human primate evidence supporting them [42].

The convex utility model is identical to the concave utility model (Eqs 16–18), with only one difference: the parameter $k_{concave}$ is replaced by a new parameter $k_{convex}$, for which $k_{convex} < 0$ holds.

**S-shaped utility.** Utilities can also be s-shaped, with different signs of curvature on both sides of a reference point [28]—concave for values below the reference point, and convex for values above. We modelled s-shaped utility functions using sign-preserving power functions

[44], which we adapted to our reward range:

$$z = \frac{r - m}{\sigma} \qquad \text{Eq 19}$$

$$U = m + \sigma \times sign(z)|z|^{k_{s-shaped}} \qquad \text{Eq 20}$$

Prediction errors are calculated as in Eq 18, updates happen according to Eq 14, choices are modeled using the softmax rule in Eq 15. The s-shaped UTIL model has free parameters $\alpha \in [0,1]$, $\beta > 0$, $k_{s-shaped} \in [0,1]$ and fixed parameters $Q_0 = 50$, $\sigma = 50$, $m = 50$.

**Inverse s-shaped utility.** The last utility function needed to complete our set is inverse s-shaped. Such utility functions are not usually considered in neuroeconomics but have been used to study the perception of numerals [44]. The inverse s-shaped utility model is identical to the s-shaped utility model, with the exception of the parameter $k_{s-shaped}$, which is substituted by $k_{inverse\ s-shaped} > 1$.

**Different learning rates for positive and negative prediction errors.** It has been shown that learning from positive outcomes can differ from learning from negative outcomes [29]. We model this by letting the learning rate depend on the sign of the prediction error (computed according to Eq 13). The update rules then become:

$$\alpha(\delta) = \begin{cases} \alpha_+ \ if \ \delta > 0 \\ \alpha_- \ if \ \delta < 0 \end{cases} \qquad \text{Eq 21}$$

$$\Delta Q_{chosen} = \alpha(\delta_{outcome}) \times \delta_{outcome} \qquad \text{Eq 22}$$

Choices are modeled using the softmax rule Eq 15. The pos-neg RATES model has free parameters $\alpha_+ \in [0,1]$, $\alpha_- \in [0,1]$, $\beta > 0$, and fixed parameters $Q_0 = 50$.

**Different learning rates for noisy and safe stimuli.** Similarly, it has been shown that the statistics of the reward distribution can have an effect on the learning rate [45,46]. In our task, normative theory [47] suggests that the learning rate should depend on the variance of the signal that is being learned. We model this by allowing different learning rates for different levels of noise. The update equations become:

$$\alpha_i = \begin{cases} \alpha_{risky} \ if \ i \in \{1,3\} \\ \alpha_{safe} \ if \ i \in \{2,4\} \end{cases} \qquad \text{Eq 23}$$

$$\Delta Q_{chosen} = \alpha_{chosen} \times \delta_{outcome} \qquad \text{Eq 24}$$

Prediction errors are computed according to Eq 13, choices are modeled using the softmax rule in Eq 15. The pos-neg RATES model has free parameters $\alpha_{risky} \in [0,1]$, $\alpha_{safe} \in [0,1]$, $\beta > 0$, and fixed parameters $Q_0 = 50$.

**PEIRS.** Our model of prediction error induced risk-seeking is based on recent models of the basal ganglia [8,9]. The relevant equations have been derived in section Results/Theory. They describe how the average $Q$ and the spread $S$ of a reward signal can be tracked. Here, $S_i$ denotes the spread of the rewards received for choosing stimulus $i$. The update equations for value are given as:

$$\Delta Q_{chosen} = \alpha_Q \times \delta_{outcome} \qquad \text{Eq 25}$$

With the outcome prediction error from Eq 13. The update equation for spread is given as:

$$\Delta S_{chosen} = \alpha_S \times (|\delta_{outcome}| - S_i)$$

Eq 26

The choice rule is a variant of the softmax rule in Eq 15:

$$p_i = \frac{exp(\beta(Q_i + tanh(\omega\delta_{stimulus})S_i))}{\sum_j exp(\beta(Q_j + tanh(\omega\delta_{stimulus})S_j))}$$

Eq 27

Here, spreads can contribute to decisions. The impact of spread on choice probability is gated by the stimulus prediction error:

$$\delta_{stimulus} = \frac{1}{2}\left(Q_{option1} + Q_{option2}\right) - \frac{1}{4}\sum_i Q_i$$

Eq 28

The PEIRS model has free parameters $\alpha_Q \in [0,1]$, $\alpha_S \in [0,1]$, $\beta > 0$, $\omega$, $S_0 > 0$ and fixed parameters $Q_0 = 50$. The initial spread estimate $S_0$ was allowed to vary since participants were not given any prior information about the spread magnitude. Initial estimates might thus have differed strongly across individuals.

**PIRS.** In another variant of the PEIRS model predictions induce risk-seeking. It differs from PEIRS only in how the stimulus prediction error is computed:

$$\delta_{stimulus} = \frac{1}{2}\left(Q_{option1} + Q_{option2}\right)$$

Eq 29

The above formula is used instead of Eq 28. Updates and choices are governed by Eqs 25, 26 and 27. The PIRS model has free parameters $\alpha_Q \in [0,1]$, $\alpha_S \in [0,1]$, $\beta > 0$, $\omega$, $S_0 > 0$ and fixed parameters $Q_0 = 50$.

**OEIRS.** If risk-preferences are associated with stimulus prediction errors, they might also be associated with outcome prediction errors from the previous trial. This is the assumption of the OEIRS model, which differs from the PEIRS model in that the outcome prediction error $\delta_{outcome}$ (computed according to Eq 13) is substituted for $\delta_{stimulus}$ in Eq 27. Otherwise, OEIRS is identical to PEIRS and PIRS.

**Scaled prediction errors.** Several studies have indicated that dopamine signaling (and hence reward prediction errors) might adapt to the magnitude of reward fluctuations [31–33]. It is thought that the adaptation is achieved through scaling reward prediction errors relative to the perceived reward variability, which is tracked alongside the average reward. Such scaling might affect stimulus-specific learning speeds and could hence introduce risk-related choice biases.

Here, we model the scaled prediction error hypothesis by implementing scaled prediction errors on top of the model of [8]. The model is then defined as

$$\tilde{\delta}_{outcome} = \frac{r - Q_{chosen}}{S_{chosen}}$$

Eq 30

$$\Delta Q_{chosen} = \alpha_Q \times \tilde{\delta}_{outcome}$$

Eq 31

$$\Delta S_{chosen} = \alpha_S \times (|\tilde{\delta}_{outcome}| - 1)$$

Eq 32

Choices are modeled using the softmax rule Eq 15. The scaled prediction error model has free parameters $\alpha_Q \in [0,1]$, $\alpha_S \in [0,1]$, $S_0 > 0$, $\beta > 0$, and fixed parameters $Q_0 = 50$.

**Attention model.**   Finally, we considered the possibility that the effects we observe might be caused by attention mechanisms—surprising outcomes (i.e., rewards that cause a high absolute prediction error) might cause subjects to be more attentive to the outcome and memorize it more thoroughly.

We modelled this as a RW learner with an additional surprise-related factor that gates learning:

$$\Delta Q_{chosen} = \alpha \times |\delta_{outcome}|^{k_{attention}} \times \delta_{outcome} \qquad \text{Eq 33}$$

Choices are modeled using the softmax rule Eq 15. The attention model has free parameters $\alpha \in [0,1]$, $k_{attention}$, $\beta > 0$, and fixed parameters $Q_0 = 50$.

**Parameter transformations and priors.**   We used exponential and sigmoid transformations to constrain the parameters to their appropriate ranges. Priors were specified as multivariate normal distributions over the untransformed parameters. All but diagonal elements of the covariance matrices of those normal distributions were set to zero. Hence, the prior distributions could be factorized into univariate normal distributions (one for each parameter). Below, we provide the statistics of those prior distributions. For parameters that occur in more than one model (such as learning rate $\alpha$), we used the same priors across models. The notation $X \sim N(\mu, \sigma)$ means that $X$ has a normal distribution with mean $\mu$ and variance $\sigma$.

$$logit(\alpha) \sim N(-1, 2)$$

$$log(\beta) \sim N(-2, 2)$$

$$log(k_{concave}) \sim N(-3, 4)$$

$$log(-k_{convex}) \sim N(-3, 4)$$

$$logit(k_{s-shaped}) \sim N(3, 4)$$

$$log(k_{inverses-shaped} - 1) \sim N(-3, 4)$$

$$logit(\alpha_{+}) \sim N(-1, 2)$$

$$logit(\alpha_{-}) \sim N(-1, 2)$$

$$logit(\alpha_{risky}) \sim N(-1, 2)$$

$$logit(\alpha_{safe}) \sim N(-1, 2)$$

$$logit(\alpha_{S}) \sim N(-1, 2)$$

$$\omega \sim N(0, 20)$$

$$log(S_0) \sim N(2, 2)$$

### Fitting & simulation

Fits were performed using the VBA toolbox [48]. This toolbox implements a Variational Bayes scheme. It takes a set of measurements, a generative probabilistic model that describes how the measurements arise (which usually contains some latent, i.e., unobserved, variables) and prior distributions over the model parameters as input, and outputs among other things an approximate posterior distribution over model parameters, an approximate posterior distribution over the latent variables, and fit statistics such as the BIC. We estimated parameters and latent variables (such as values and prediction errors) using the mean of the posterior distributions over parameters from the toolbox outputs.

Simulations were performed with parameters taken from fits: for each model, a fit provided us with 27 parameter sets corresponding to the 27 participants. Using those parameter sets we simulated our task and generated datasets of the same size as the dataset we obtained in our experiment. We repeated this 1000 times to obtain stable distributions.

We extracted 27000 pairs of aggregate risk preferences and risk traces (risk preferences as a function of trial number) per model from the simulated datasets (27 simulated participants in each of 1000 simulated experiments).

For Fig 3A, we used a kernel smoothing function to estimate the probability density underlying the distribution of risk preferences (MATLAB's *ksdensity* function). We then computed and visualized isolines of those estimated probability densities using MATLAB's *contour* function (Fig 3A).

For Fig 3B, we first split the risk preferences between conditions. We then averaged the traces over repetitions of the experiment for each simulated participant, and finally over participants. For display, the averaged simulated traces were smoothed with a 20-point moving average filter.

For Fig 3C, for each model we first computed the average difference in risk preference between conditions for each simulated participant in each simulated experiment, yielding 27000 differences in risk preference per model. We then averaged those across experiments, obtaining 27 differences in risk preferences per model. Those distributions were compared to the empirical distribution of mean difference in risk preference between conditions, which has 27 data points as well (one for each participant).

## Supporting information

**S1 Text. File containing details of data analysis methods, additional analyses and Fig A-F.** (DOCX)

## Author Contributions

**Conceptualization:** Moritz Moeller, Jan Grohn, Sanjay Manohar, Rafal Bogacz.

**Investigation:** Moritz Moeller, Jan Grohn.

**Supervision:** Sanjay Manohar, Rafal Bogacz.

**Writing – original draft:** Moritz Moeller, Jan Grohn.

**Writing – review & editing:** Moritz Moeller, Jan Grohn, Sanjay Manohar, Rafal Bogacz.

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
