## [Decision Letter · Decision Letter 0]

18 Apr 2021

Dear Dr. Bogacz,

Thank you very much for submitting your manuscript "A Behavioral Association Between Prediction Errors and Risk-Seeking: Theory and Evidence" for consideration at PLOS Computational Biology.

As with all papers reviewed by the journal, your manuscript was reviewed by members of the editorial board and by several independent reviewers. In light of the reviews (below this email), we would like to invite the resubmission of a significantly-revised version that takes into account the reviewers' comments.

We cannot make any decision about publication until we have seen the revised manuscript and your response to the reviewers' comments. Your revised manuscript is also likely to be sent to reviewers for further evaluation.

Sincerely,

Alireza Soltani

Associate Editor

PLOS Computational Biology

Samuel Gershman

Deputy Editor

PLOS Computational Biology

Reviewer's Responses to Questions

**Comments to the Authors:**

Reviewer #1: Moeller et al attempt to unify two classic functions attributed to dopamine neurons – reward prediction errors and risk preferences. Building off of beautiful and rigorous computational work by the same authors (Mikhael and Bogacz 2016; Moller and Bogacz 2019), the authors develop a neurally-inspired theory that attempts to unify these two functions. Motivated by the theory, they develop a two-alternative forced choice task and find behavioral and pupillometric evidence in support of their theory.

Overall, I find the task design and behavioral analysis interesting, compelling, and non-trivial. However, I remain rather unconvinced by the theory, the computational modeling, and the pupillometric data, and I believe a substantial amount of work would be needed to make them compelling.

Major Comments

I find the extension of the theory in this manuscript a bit arbitrary. The theory is developed from the perspective of basal ganglia circuitry and the authors posit the existence of action channels (Ai) modulated by direct pathway excitation (Gi) and indirect pathway inhibition (Ni). The activation of these channels is proportional to the probability of choosing option i. The authors then develop Eq. 1 as a consequence of dopamine levels of above/below baseline. However, one can rewrite Eq. 1 as follows -

Ai=(δ+0.5)Gi + (δ-0.5)Ni

Written this way, if the system receives δ>0.5, then Ni, the indirect pathway inhibition, now increases the probability of choosing Ai. This appears counterintuitive to me, at given the description of the system. It seems to me that the equation written in line 177, Ai=Qi + δSi, is a more natural starting point as a hypothesis for how the system may function, with Eq. 1 as a consequence. However, if this is a starting point, I do not see a priori why this particular functional form was chosen and not any other (for example, Ai=Qi - δSi, to allow for risk aversion for gains and risk preference for losses). In other words, there seems to be enough flexibility with the development of the theory to allow it to fit the empirical data at hand. Unless I am missing something, this flexibility greatly limits the ability to interpret the behavioral results as a natural consequence of the basal-ganglia-inspired theory.

The Prediction Errors Induce Risk Seeking (PEIRS) model is presented as the most parsimonious model for describing how participants modulate risk seeking as a function of stimulus prediction errors. I agree with the authors that this seems to be the only model that could capture both the risk preference when δ_stimulus>0 and the risk aversion when δ_stimulus<0. However, the core assumption of this model, that δ_stimulus=〖mean〗_shown (Q)-〖mean〗_all (Q), makes the prediction that there should be negative prediction errors for reward-predicting stimuli. There are numerous examples in the literature (Tobler, Fiorillo, Schultz 2005 as one compelling example, 10.1126/science.1105370) that show that dopamine neurons increase firing in anticipation of reward-predicting cues, even if less than the average over all stimuli. Given the neurally-inspired origins of the authors’ theory, it seems that PEIRS likely must be rejected on the basis of extant neural recordings.

The authors present the Predictions Induce Risk Seeking (PIRS) model as an alternative to PEIRS. The δ_stimulus for PIRS is strictly positive in this task (lines 215-216), which accords well with neural data. This model is able to capture the behavioral data in Figure 2C, and shows stronger risk preference for the high-value stimuli than for the low-value stimuli. However, this model predicts risk-seeking for all δ_stimulus>0, which seems to run directly counter to the behavioral results shown in Figure 2B. Given a clear behavioral effect, I would argue that these data falsify the model.

Why only simulate the risk preference data in Figure 2C? The authors mention several times that a key advantage of their approach is to simulate trial-by-trial data (e.g. lines 581-583) yet they only show the summary statistic for risk preferences. The simulations would strongly benefit from trial-by-trial analysis to show that the best models also capture the learning effects in Figures 2A,B. As mentioned above, this analysis would have likely falsified PIRS.

Assuming that the computational form of PEIRS is correct (though, as I mentioned above, I do not think the neural implementation would correspond to real neural data), the model makes the prediction that when δ_stimulus=0, there should be no risk preference. This is a clear prediction of the model that should be tested.

Support for the PEIRS model would be strengthened dramatically if the authors can show that risk preferences for the exact same stimuli vary as a function of δ_stimulus. As one example, imagine a task with six stimuli – risky-high, safe-high, risky-medium, safe-medium, risky-low, safe-low. On a particular trial, subjects can be cued that the possible set of stimuli for that trial will be drawn from risky-high, safe-high, risky-medium, safe-medium. This would set the baseline for comparison. Then, subjects would be given risky-medium and safe-medium as options on that trial. This would elicit δ_stimulus<0, and the theory predicts that participants would choose safe-medium. If, however, the set of possible stimuli for that trial were risky-medium, safe-medium, risky-low, and safe-low, and subjects were given risky-medium and safe-medium as options, this would elicit δ_stimulus>0, and the theory would predict that participants should choose risky-medium. Though this particular task variant may not be feasible, a task that tests this idea would strongly support the idea that δ_stimulus modulates risk preferences.

Given the trial-by-trial nature of the PEIRS model, do the authors have the power to assess trial-by-trial changes in risk preference as a function of updates to S? In other words, how do risk preferences evolve following large updates to S vs small updates (or decreases) to S? Though I suspect the authors may be underpowered for this analysis, this is another prediction of the theory.

Regarding the pupillometric data, I have two questions/concerns. First, the authors should include other relevant regressors in their models (e.g. stimulus identity, stimulus chosen, relevant latent variables) and see how much their results change. Second, shouldn’t the pupil-behavior comparison for Figure 4C be a comparison between pupil response and strength of risk preference? I would expect those participants with the strongest pupillometric responses (and therefore, the strongest encoding of δ_stimulus) would show the strongest risk preferences (and risk preference asymmetry, as according to the model).

Minor comments

Eq. 3 appears miswritten – as is, S would always increase. I imagine the authors meant to put Eq. 12 in its place.

Figure 2C – at what timepoints are the risk preferences being measured? I would assume the latter half of the session, when choices and risk preferences have stabilized, though this needs to be reported.

The predictions in lines 250-251 do not match with order of predictions in lines 289, 301.

In line 308, the authors state “This is consistent with our theory: the first effect rests on more assumptions than the second.” It is not clear how this follows – an effect resting on more assumptions does not guarantee a weaker effect size.

Eq. 10 - δ_coutcome should be δ_outcome

In line 662, authors state “Utilities can also be s-shaped, with different curvatures on both sides of a reference point.” However, the proposed utility function is symmetric on both sides of 0.

In line 666: U = m + s, should be written as U = m + sigma

The following statements are far too strong for what is shown, as showing a significant association does not prove the existence of that signal (e.g., unmeasured confounders, etc…)

Line 436: “we prove the occurrence of stimulus prediction errors by extracting the corresponding pupil response.”

Lien 475: “This confirms the existence of a stimulus prediction error that occurs after stimulus onset, as hypothesized.”

In estimating the pupillometric stimulus prediction error signal, the authors censored the pupil recordings after reward presentation. It seems that the pupil signal contains time periods corresponding to choices. What is the effect of choice on the pupil signal and could that affect the interpretation?

Describe the cluster-based permutation test in the methods.

Q_0 is a fixed parameter, set to 50. How would model fits change if Q_0 was a free parameter or if it was set to 0? S_0 is set as a free parameter, so it seems worthwhile exploring the effects of Q_0 being free as well.

Figure 4C needs units.

Reviewer #2: In the manuscript titled “A Behavioral Association Between Prediction Errors and Risk-Seeking: Theory and Evidence”, Moeller and Colleagues use computational approaches alongside behavioral experiments to investigate the effect reward prediction errors (RPEs) have on risk preferences. Specifically, authors hypothesize that when learning about risky options (e.g., options with reward magnitudes that are drawn from different distributions), choice behavior is modulated by stimuli prediction errors (how much the average value of the presented stimuli differ from the average value of all stimuli). Authors then used a simple arm bandit task to investigate the predictions of their hypothesis.

I find the question proposed in this paper is of value. However, I have major concerns about the strength of the results presented and the analysis performed, which needs to be addressed. Additionally, the writing needs to improve dramatically, the paper is not very well written and difficult to read (some suggestions have been provided below). Please find my comments/concerns:

Major concerns:

1) Page 2, line 18: Authors don’t talk about their hypothesis/theory until line 21 and even then, it is not clear what their hypothesis is. Please revise the abstract and clarify what is being tested/hypothesized.

2) Authors only include concave and s-shaped utility functions in their analysis. However, it seems to me that including a convex or inverted s-shaped utility function can explain parts of the reported results. Convex utility functions have been reported in non-human primates ([1]).

3) Similarly, models that allow for scaling of RPEs ([2-3]), may be able to explain the observed behaviors and should be tested.

4) Authors use the first 60 trials when calculating the correlation between pupil dilation and outcome prediction errors, while they use last 60 trials for calculating the correlation between pupil dilation and stimulus prediction error. If the fact that learning has been accomplished by 60 trials means that the RPE doesn’t exists (which is not a correct assumption), then it doesn’t make sense to include final 60 trials for outcome prediction error. What will happen if all trials are included in both analysis?

5) Page 22, Fig. 4C: it seems that there are two types of subjects, those whose value for (BIC_rw – BIC_peirs) are negative (adopted RW model) and those whose value for (BIC_rw – BIC_peirs) are positive (adopted PEIRS model). Surprisingly, the majority of subjects’ behavior is better explained by RW model. What will happen if percentage of subjects with smaller values of BIC are plotted in Fig. 3C, D. This figure suggests that among all subjects (30 of them), the behavior of only 7 of them was better explained by PEIRS model, which is driving the results in Fig. 3C, D, as well as the correlation in Fig. 4C.

6) Authors use (BIC_rw – BIC_peirs) to quantify the prediction error induced risk preferences. Do authors have any evidence supporting this choice? For example, is there any correlation between the value of \\gamma and the difference in BIC values? Then why not use \\gamma values? Please clarify.

7) Comparison of likelihood of proposed models is used to provide strong evidence that PEIRS/PIRS variant of models can best explain the subjects’ behavior. This begs the question of whether the models are well identifiable in this task. To test this, authors should simulate each model with a range of parameters, and fit the simulated behavior with each model, then show that each model fits better data simulated by the same model. This procedure also allows them to verify that the parameters are identifiable by the fitting procedure.

Minor concerns:

1) Page 3, line 51-56: Please revise this sentence, it is very difficult to understand.

2) Please provide a supplementary figure with average choice probability values for the other one condition (different).

3) The performance cut-off value seems arbitrary, why 70% was chosen (Fig. S1 caption, however, says 65%)? Please clarify.

4) Please fix the following typos:

Page 22, line 458: …(trials 61-120).similar to A) (This time interval is not similar to A)

Page 31, Models: Please add equation number to all formulas.

Page 2, line 36: … is precisely what our theory predicts …

Page 12, line 244: … this also contributes to a higher (in) learning …

Page 12, line 246: Relaxing (now) our assumption

Page 31 line 627: … preferences emerged faster than risk preferences …

Page 33, line 678: Different learning rates for noisy and safe stimuli

Page 34, line 692: … the spread ()of a reward signal …

Page 34, line 693: … denotes (the spread) of the spread of the rewards

Page 38, line 775: … against a trial-by-trial estimate …

5) Page 25, line 499: No proper motivation is given for the study. Please revise.

6) Page 25, line 504: … determines its effect on risk preferences.

Using this/that/it/such is not very informative and should be avoided (the issue persists throughout discussion section (e.g., line 517 and etc.)). Please revise.

7) Page 26, line 527: This information (exact numbers) should be provided in the results section.

8) Page 28, line 587-8: This sentence is not written in a scientific tone. Please revise.

9) Page 31, line 626-627: This sentence belongs to results not methods section. Please remove it.

References:

[1]. Stauffer, W. R., Lak, A., Bossaerts, P., & Schultz, W. (2015). Economic choices reveal probability distortion in macaque monkeys. Journal of Neuroscience, 35(7), 3146-3154.

[2]. Diederen, K. M., & Schultz, W. (2015). Scaling prediction errors to reward variability benefits error-driven learning in humans. Journal of Neurophysiology, 114(3), 1628-1640.

[3]. Diederen, K. M., Spencer, T., Vestergaard, M. D., Fletcher, P. C., & Schultz, W. (2016). Adaptive prediction error coding in the human midbrain and striatum facilitates behavioral adaptation and learning efficiency. Neuron, 90(5), 1127-1138.

Reviewer #3: The main finding of this paper is that stimulus RPEs (quantified as mean value of presented options relative to either (a) mean of all options or (b) 0) drive an increase in risk seeking. The finding is nicely grounded in well-established models of basal ganglia function.

Short paper summary:

Theory: After stimulus +RPE, people would ignore all below average rewards; this pulls the value of risky stimuli higher, away from their respective means

Main result: people are more risk seeking in the +RPE condition

Secondary result: people who have more RPE induced risk seeking (measured as degree to which PEIRS model wins over RW) show stronger pupillary response

Comments/suggestions/questions:

* The behavioral findings are interesting and clean, and the hypothesis clear

* Could the effects be explained by attentionally-mediated higher learning rates in the high-risk condition? This would be similar to the RATES model (at least as I understood it), but with a dynamic rather than static individual-level learning rate; one possible way to test this: compute “empirical” learning rates from pupillary response, or "theoretical" learning rates from Eqs. 4 and 5

* Please write out at least the complete winning models (PEIRS & PIRS) in their own separate box; as is, it’s quite difficult to follow how different models build on one another

* I didn't follow from the exposition what the authors expect the role of -RPEs to be for the both-low condition; and were trials of different conditions blocked? Would be good to state this upfront in the task figure and on lines 145-149

* Please be more clear early in the text that you mean stimulus prediction errors when you lay out the theory; on a first read I didn’t catch until the discussion that the paper was attributing the risk effect to “stimulus prediction errors”

* Re: discussion, it’s hard to make inferences about dopamine from a behavioral task with eye-tracking. For instance, it’s not clear whether the pupillary surprise signal is noradrenergically or dopaminergically modulated (Takeuchi … Morris 2016). Given the strong pupil effects authors observe, it might be that the surprise effect is driven by NE, which has classically been associated with the pupillary response, and with an unsigned “surprise” signal

* Relatedly, could the effect be mediated by memory effects? Recent work has linked surprise to better memory, irrespective of the sign of RPEs (Rouhani … Niv 2018)

* It would be helpful to place this work in the context of the Yu and Dayan expected vs. unexpected uncertainty model

Thanks for an interesting read!

**Have all data underlying the figures and results presented in the manuscript been provided?**

Reviewer #1: Yes

Reviewer #2: None

PLOS authors have the option to publish the peer review history of their article (what does this mean?). If published, this will include your full peer review and any attached files.

Reviewer #1: No

Reviewer #2: No

Reviewer #3: No

**Have the authors made all data and (if applicable) computational code underlying the findings in their manuscript fully available?**

Reviewer #3: Yes
---

## [Decision Letter · Decision Letter 1]

23 Jun 2021

Dear Dr. Bogacz,

We are pleased to inform you that your manuscript 'An Association Between Prediction Errors and Risk-Seeking: Theory and Behavioral Evidence' has been provisionally accepted for publication in PLOS Computational Biology.

Best regards,

Alireza Soltani

Associate Editor

PLOS Computational Biology

Samuel Gershman

Deputy Editor

PLOS Computational Biology

Reviewer's Responses to Questions

**Comments to the Authors:**

Reviewer #1: Moeller and colleagues have provided a thoroughly and thoughtfully revised, simplified, and – in my view – significantly improved manuscript. In sum, this manuscript is ready for publication.

The development of the theory is much clearer in the current manuscript. By referencing the relevant starting point in Moller, Bogacz (2019), the reader can easily follow the extension and understand how risk preference/aversion arises from the theoretical formulation. The theory now stands as a highlight of the paper. The additional neural data supporting PEIRS (Wang et al 2021) is well-received and the relevant discussion highlights the important points (that stimulus presentation itself is associated with a burst, separate from the stimulus prediction error). I applaud the removal of PIRS and the pupillometric data from the main focus of the manuscript. This greatly streamlines the point and makes the behavioral findings stand more sharply in focus. The new trial-by-trial simulated data is a welcome addition and makes it clear that PEIRS is the superior model (for example, pos-neg RATES has a similar BIC but clearly fails to capture trial-by-trial risk preferences). Regarding the additional experiments, I believe the manuscript and theory would be improved if the authors could experimentally validate the prediction that d_stimulus = 0 yields no risk preference (rather than interpolating via logistic regression). However, I believe with the more streamlined manuscript, this is not necessary for publication. I thank the authors for tactfully assuaging my concerns and I believe the manuscript has improved as a result.

Reviewer #2: I thank the authors for addressing my concerns. I only have a comment;

Regarding major point 4) Authors use 'learning has finished' to refer to the fact that 'behavior/performance has reached its steady state'. Their term is confusing.

Reviewer #3: The authors have satisfactorily addressed my concerns and I have no further comments.

**Have the authors made all data and (if applicable) computational code underlying the findings in their manuscript fully available?**

Reviewer #1: Yes

Reviewer #2: None

Reviewer #3: None

PLOS authors have the option to publish the peer review history of their article (what does this mean?). If published, this will include your full peer review and any attached files.

Reviewer #1: No

Reviewer #2: No

Reviewer #3: No

---

## [Editor Report · Acceptance letter]

12 Jul 2021

PCOMPBIOL-D-21-00283R1 

An Association Between Prediction Errors and Risk-Seeking: Theory and Behavioral Evidence

Dear Dr Bogacz,

I am pleased to inform you that your manuscript has been formally accepted for publication in PLOS Computational Biology. Your manuscript is now with our production department and you will be notified of the publication date in due course.

With kind regards,

Andrea Szabo
